# ADDITIVE POWERS-OF-TWO QUANTIZATION: AN EFFICIENT NON-UNIFORM DISCRETIZATION FOR NEURAL NETWORKS

**Yuhang Li** [†*]**, Xin Dong** [§*]**, Wei Wang** [†]
[†]National University of Singapore, [§]Harvard University
`loafyuhang@gmail.com, xindong@g.harvard.edu, wangwei@comp.nus.edu.sg`

## ABSTRACT

We propose **A**dditive **P**owers-**o**f-**T**wo (APoT) quantization, an efficient non-uniform quantization scheme for the bell-shaped and long-tailed distribution of weights and activations in neural networks. By constraining all quantization levels as the sum of Powers-of-Two terms, APoT quantization enjoys high computational efficiency and a good match with the distribution of weights. A simple reparameterization of the clipping function is applied to generate a better-defined gradient for learning the clipping threshold. Moreover, weight normalization is presented to refine the distribution of weights to make the training more stable and consistent. Experimental results show that our proposed method outperforms state-of-the-art methods, and is even competitive with the full-precision models, demonstrating the effectiveness of our proposed APoT quantization. For example, our 4-bit quantized ResNet-50 on ImageNet achieves 76.6% top-1 accuracy without bells and whistles; meanwhile, our model is capable to decrease 22% computational cost compared with the uniformly quantized counterpart. [1]

## 1 INTRODUCTION

Deep Neural Networks (DNNs) have made a significant improvement for various real-world applications. However, the huge memory and computational cost impede the mass deployment of DNNs, e.g., on resource-constrained devices. To reduce memory footprint and computational burden, several model compression methods such as quantization (Zhou et al., 2016), pruning (Han et al., 2015) and low-rank decomposition (Denil et al., 2013) have been widely explored.

In this paper, we focus on the neural network quantization for efficient inference. Two operations are involved in the quantization process, namely clipping and projection. The clipping operation sets a full precision number to the range boundary if it is outside of the range; the projection operation maps each number (after clipping) into a predefined quantization level (a fixed number). We can see that both operations incur information loss. A good quantization method should resolve the two following questions/challenges, which correspond to two contradictions respectively.

*How to determine the optimal clipping threshold to balance clipping range and projection resolution?* The resolution indicates the interval between two quantization levels; the smaller the interval, the higher the resolution. The first contradiction is that given a fixed number of bits to represent weights, the range and resolution are inversely proportional. For example, a larger range can clip fewer weights; however, the resolution becomes lower and thus damage the projection. Note that slipshod clipping of outliers can jeopardize the network a lot (Zhao et al., 2019) although they may only take 1-2% of all weights in one layer. Previous works have tried either pre-defined (Cai et al., 2017; Zhou et al., 2016) or trainable (Choi et al., 2018b) clipping thresholds, but how to find the optimal threshold during training automatically is still not resolved.

---

[*]Equal Contribution.     Y. L. completed this work during his internship at NUS.
[1]Code is available at `https://github.com/yhhhli/APoT_Quantization`.

*How to design quantization levels with consideration for both the computational efficiency and the distribution of weights?* Most of the existing quantization approaches (Cai et al., 2017; Gong et al., 2019) use uniform quantization although non-uniform quantization can usually achieve better accuracy (Zhu et al., 2016). The reason is that projection against uniform quantization levels are much more hardware-friendly (Zhou et al., 2016). However, empirical study (Han et al., 2015) has shown that weights in a layer of DNN follow a bell-shaped and long-tailed distribution instead of a uniform distribution (as shown in the right figure). In other words, a fair percentage of weights concentrate around the mean (peak area); and a few weights are of relatively high magnitude and out of the quantization range (called outliers). Such distribution also exists in activations (Miyashita et al., 2016). The second contradiction is: considering the bell-shaped distribution of weight, it is well-motivated to assign higher resolution (i.e. smaller quantization interval) around the mean; however, such non-uniform quantization levels will introduce

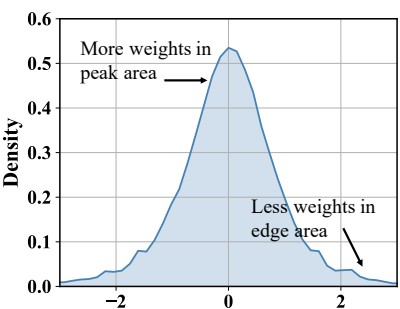

Figure 1: Density of weights in ResNet-18

high computational overhead. Powers-of-Two quantization levels (Miyashita et al., 2016; Zhou et al., 2017) are then proposed because of its cheap multiplication implemented by *shift* operations on hardware, and super high resolution around the mean. However, the vanilla powers-of-two quantization method only increases the resolution near the mean and ignores other regions at all when the bit-width is increased. Consequently, it assigns inordinate quantization levels for a tiny range around the mean. To this end, we propose additive Powers-of-Two (APoT) quantization to resolve these two contradictions, our contribution can be listed as follows:

1. We introduce the APoT quantization scheme for the weights and activations of DNNs. APoT is a non-uniform quantization scheme, in which the quantization levels is a sum of several PoT terms and can adapt well to the bell-shaped distribution of weights. APoT quantization enjoys an approximate $2\times$ multiplication speed-up compared with uniform quantization on both generic and specific hardware.
2. We propose a Reparameterized Clipping Function (RCF) that can compute a more accurate gradient for the clipping threshold and thus facilitate the optimization of the clipping threshold. We also introduce weight normalization for neural network quantization. Normalized weights in the forward pass are more stable and consistent for clipping and projection.
3. Experimental results show that our proposed method outperforms state-of-the-art methods, and is even competitive with the full-precision implementation with higher computational efficiency. Specifically, our 4-bit quantized ResNet-50 on ImageNet achieve 76.6% Top-1 and 93.1% Top-5 accuracy. Compared with uniform quantization, our method can decrease 22% computational cost, demonstrating the proposed algorithm is hardware-friendly.

## 2 METHODOLOGY

### 2.1 PRELIMINARIES

Suppose kernels in a convolutional layer are represented by a 4D tensor $\mathcal{W} \in \mathbb{R}^{C_{out} \times C_{in} \times K \times K}$, where $C_{out}$ and $C_{in}$ are the number of output and input channels respectively, and $K$ is the kernel size. We denote the quantization of the weights as

$$\hat{\mathcal{W}} = \Pi_{\mathcal{Q}(\alpha,b)} \lfloor \mathcal{W}, \alpha \rceil, \tag{1}$$

where $\alpha$ is the clipping threshold and the clipping function $\lfloor \cdot, \alpha \rceil$ clips weights into $[-\alpha, \alpha]$. After clipping, each element of $\mathcal{W}$ is projected by $\Pi(\cdot)$ onto the quantization levels. We denote $\mathcal{Q}(\alpha, b)$ for a set of quantization levels, where $b$ is the bit-width. For uniform quantization, the quantization levels are defined as

$$\mathcal{Q}^u(\alpha, b) = \alpha \times \{0, \frac{\pm 1}{2^{b-1}-1}, \frac{\pm 2}{2^{b-1}-1}, \frac{\pm 3}{2^{b-1}-1}, \dots, \pm 1\}. \tag{2}$$

For every floating-point number, uniform quantization maps it to a $b$-bit fixed-point representation (quantization levels) in $\mathcal{Q}^u(\alpha, b)$. Note that $\alpha$ is stored separately as a full-precision floating-point

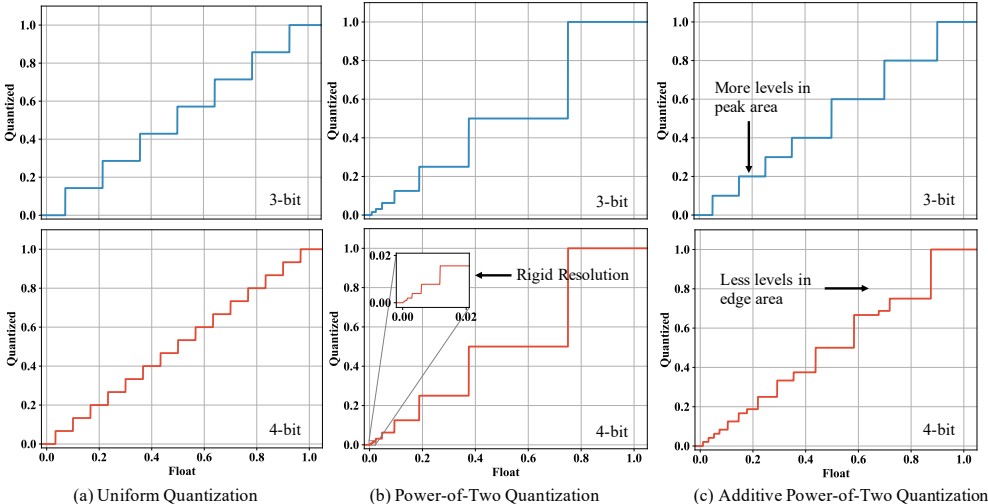

Figure 2: Quantization of unsigned data to 3-bit or 4-bit ($\alpha = 1.0$) using three different quantization levels. APoT quantization has a more reasonable resolution assignment and it does not suffer from the rigid resolution.

number for each whole $\mathcal{W}$. Convolution is done against the quantization levels first and the results are then multiplied by $\alpha$. Arithmetical computation, e.g., convolution, can be implemented using low-precision fixed point operations on hardware, which are substantially cheaper than their floating-point contradictory (Goldberg, 1991). Nevertheless, uniform quantization does not match the distribution of weights (and activations), which is typically bell-shaped (Han et al., 2015). A straightforward solution is to assign more quantization levels (higher resolution) for the peak of the distribution and fewer levels (lower resolution) for the tails. However, it is difficult to implement the arithmetical operations for the non-uniform quantization levels efficiently.

## 2.2 ADDITIVE POWERS-OF-TWO QUANTIZATION

To solve the contradiction between non-uniform resolution and hardware efficiency, Powers-of-Two (PoT) quantization (Miyashita et al., 2016; Zhou et al., 2017) is proposed by constraining quantization levels to be powers-of-two values or zero, i.e.,

$$\mathcal{Q}^p(\alpha, b) = \alpha \times \{0, \pm 2^{-2^{b-1}+1}, \pm 2^{-2^{b-1}+2}, ..., \pm 2^{-1}, \pm 1\}. \qquad (3)$$

Apparently, as a non-uniform quantizer, PoT has a higher resolution for the value range with denser weights because of its exponential property. Furthermore, multiplication between a Powers-of-two number $2^x$ and the other operand $r$ can be implemented by bit-wise shift instead of bulky digital multipliers, i.e.,

$$2^x r = \begin{cases} r & \text{if } x = 0 \\ r << x & \text{if } x > 0, \\ r >> x & \text{if } x < 0 \end{cases} \qquad (4)$$

where $>>$ denotes the right *shift* operation and is computationally cheap, which only takes 1 clock cycle in modern CPU architectures.

However, we find that PoT quantization does not benefit from more bits. Assume $\alpha$ is 1, as shown in Equation (3), when we increase the bit-width from $b$ to $b+1$, the interval $[-2^{-2^{b-1}+1}, 2^{-2^{b-1}+1}]$ will be split into $2^{b-1} - 1$ sub-intervals, whereas all other intervals remain unchanged. In other words, by increasing the bit-width, the resolution will increase only for $[-2^{-2^{b-1}+1}, 2^{-2^{b-1}+1}]$. We refer this phenomenon as the *rigid resolution* of PoT quantization. Take $\mathcal{Q}^p(1, 5)$ as an example, the two smallest positive levels are $2^{-15}$ and $2^{-14}$, which is excessively fine-grained. In contrast, the two largest levels are $2^{-1}$ and $2^0$, whose interval is large enough to incur high projection error for weights between $[2^{-1}, 2^0]$, e.g., 0.75. The rigid resolution is demonstrated in Figure 2(b). When we change from from 3-bit to 4-bit, all new quantization levels concentrate around 0 and thus cannot increase the model's expressiveness effectively.

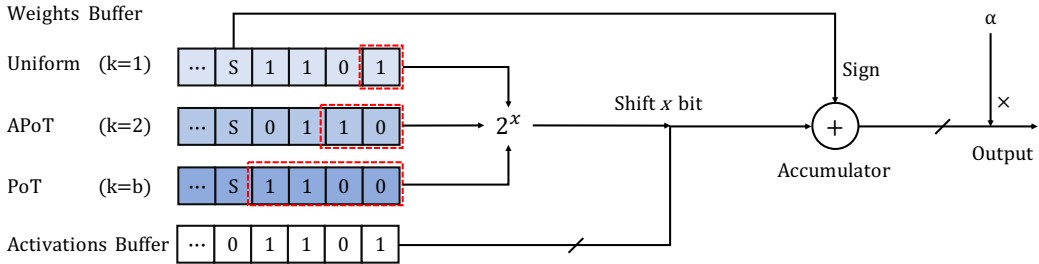

Figure 3: Hardware accelerator with different quantization schemes. When $k$ increase, weights usually has less PoT terms, thus accelerates the computation.

To tackle the *rigid resolution* problem, we propose Additive Powers-of-Two (APoT) quantization. Without loss of generality, in this section, we only consider unsigned numbers for simplicity[2]. In APoT quantization, each level is the sum of n PoT terms as shown below,

$$\mathcal{Q}^a(\alpha, kn) = \gamma \times \{ \sum_{i=0}^{n-1} p_i \} \text{ where } p_i \in \{0, \frac{1}{2^i}, \frac{1}{2^{i+n}}, ..., \frac{1}{2^{i+(2^k-2)n}}\}, \tag{5}$$

where $\gamma$ is a scaling coefficient to make sure the maximum level in $\mathcal{Q}^a$ is $\alpha$. $k$ is called the base bit-width, which is the bit-width for each additive term, and $n$ is the number of additive terms. When the bit-width $b$ and the base bit-width $k$ is set, $n$ can be calculated by $n = \frac{b}{k}$. There are $2^{kn} = 2^b$ levels in total. The number of additive terms in APoT quantization can increase with bit-width $b$, which provides a higher resolution for the non-uniform levels.

We use $b = 4$ and $k = 2$ as an example to illustrate how APoT resolves the *rigid resolution* problem. For this example, we have $p_0 \in \{0, 2^0, 2^{-2}, 2^{-4}\}$, $p_1 \in \{0, 2^{-1}, 2^{-3}, 2^{-5}\}$, $\gamma = 2\alpha/3$, and $\mathcal{Q}^a(\alpha, kn) = \{\gamma \times (p_0 + p_1)\}$ for all ($2^b = 16$) combinations of $p_0$ and $p_1$. First, we can see the smallest positive quantization level in $\mathcal{Q}^a(1, 4)$ is $2^{-4}/3$. Compared with the original PoT levels, APoT allocates quantization levels prudently for the central area. Second, APoT generates 3 new quantization levels between $2^0$ and $2^{-1}$, to properly increase the resolution. In Figure 2, the second row compares the 3 quantization methods using 4 bits for range [0, 1]. APoT quantization has a reasonable distribution of quantization levels, with more levels in the peak area (near 0) and relatively higher resolution than the vanilla PoT quantization at the tail (near 1).

**Relation to other quantization schemes.** On the one hand, the fixed-point number representations used in the uniform quantization is a special case of APoT. When $k = 1$ in Equation (5), the quantization levels is a sum of $b$ PoT terms or 0. In the fixed-point representations, each bit indicates one specific choice of the additive terms. On the other hand, when $k = b$, there is only one PoT term and $\mathcal{Q}^a(\alpha, b)$ becomes $\mathcal{Q}^p(\alpha, b)$, i.e., PoT quantization. We can conclude that when $k$ decreases, APoT levels are decomposed into more PoT terms, and the distribution of levels becomes more uniform. Our experiments use $k = 2$, which is an intermediate choice between the uniform case ($k = 1$) and the vanilla PoT case ($k = b$).

**Computation.** Multiplication for fixed-point numbers can be implemented by shifting the multiplicand (i.e., the activations) and adding the partial product. The $n$ in Equation (5) denotes the number of additive PoT terms in the multiplier (weights), and control the speed of computation. Since $n = \frac{b}{k}$, either decreasing $b$ or increasing $k$ can accelerate the multiplication. Compared with uniform quantization ($k = 1$), our method ($k = 2$) is approximately $2\times$ faster in multiplication. As for the full precision $\alpha$, it is a coefficient for all weights in a layer and can be multiplied only once after the multiply-accumulate operation is finished. Figure 3 shows the hardware accelerator, the weights buffer takes $k$-bit as a PoT term and *shift-adds* the activations.

**Generalizing to $2n + 1$ bits.** When $k = 2$, APoT quantization can only leverages $2n$-bit width for quantization. To deal with $2n + 1$-bit quantization, we choose to add $n + 1$ PoT terms, one of which

---

[2]To extend the solution for the signed number, we only need to add 1 more bit for the sign.

only contains 2 levels. The formulation is given by

$$\mathcal{Q}^a(\alpha, 2n+1) = \gamma \times \left\{ \sum_{i=0}^{n-1} p_i + \tilde{p} \right\} \text{ where } p_i \in \{0, \frac{1}{2^i}, \frac{1}{2^{i+n}}, \frac{1}{2^{i+2n+1}}\}, \ \tilde{p} \in \{0, \frac{1}{2^{i+2n}}\}. \quad (6)$$

Take 3-bit APoT quantization as an example, every level is a sum of one $p_0$ and one $\tilde{p}$, where $p_0 \in \{0, 2^{-1}, 2^{-2}, 2^{-4}\}$ and $\tilde{p} \in \{0, 2^{-3}\}$. The forward function is plotted in Figure 2(c).

## 2.3 REPARAMETERIZED CLIPPING FUNCTION

Besides the projection operation, the clipping operation $\lfloor \mathcal{W}, \alpha \rceil$ is also important for quantization. $\alpha$ is a threshold that determines the value range of weights in a quantized layer. Tuning the clipping threshold $\alpha$ is a key challenge because of the long-tail distribution of the weights. Particularly, if $\alpha$ is too large (e.g., the maximum absolute value of $\mathcal{W}$), $\mathcal{Q}(\alpha, b)$ would have a wide range and then the projection will lead to large error as a result of insufficient resolution for the weights in the central area; if $\alpha$ is too small, more outliers will be clipped slipshodly. Considering the distribution of weights can be complex and differs across layers and training steps, a static clipping threshold $\alpha$ for all layers is not optimal.

To jointly optimize the clipping threshold $\alpha$ and weights via SGD during training, Choi et al. (2018b) apply the Straight-Through Estimator (STE) (Bengio et al., 2013) to do the backward propagation for the projection operation. According to STE, the gradient to $\alpha$ is computed by $\frac{\partial \hat{\mathcal{W}}}{\partial \alpha} \approx \frac{\partial \lfloor \mathcal{W}, \alpha \rceil}{\partial \alpha} = \text{sign}(\mathcal{W})$ when $|\mathcal{W}| > \alpha$ otherwise 0, where the weights outside of the range cannot contribute to the gradients, which results in inaccurate gradient approximation. To provide a refined gradient for the clipping threshold, we design a Reparameterized Clipping Function (RCF) as

$$\hat{\mathcal{W}} = \alpha \Pi_{\mathcal{Q}(1,b)} \lfloor \frac{\mathcal{W}}{\alpha}, 1 \rceil. \quad (7)$$

Instead of directly clipping them to $[-\alpha, \alpha]$, RCF outputs a constant clipping range and re-scales weights back after the projection, which is mathematically equivalent to Equation (1) during forward. In backpropagation, STE is adopted for the projection operation and the gradients of $\alpha$ are calculated by

$$\frac{\partial \hat{\mathcal{W}}}{\partial \alpha} = \begin{cases} \text{sign}(\mathcal{W}) & \text{if } |\mathcal{W}| > \alpha \\ \Pi_{\mathcal{Q}(1,b)} \frac{\mathcal{W}}{\alpha} - \frac{\mathcal{W}}{\alpha} & \text{if } |\mathcal{W}| \leq \alpha \end{cases} \quad (8)$$

The detail derivation of the gradients is shown in Appendix A. Compared with the normal clipping function, RCF provides more accurate gradient signals for the optimization because both weights inside ($|\mathcal{W}| \leq \alpha$) and out of ($|\mathcal{W}| > \alpha$) the range can contribute to the gradient for the clipping threshold. Particularly, the outliers are responsible for the clipping, and the weights in $[-\alpha, \alpha]$ are for projection. Therefore, the update of $\alpha$ considers both clipping and projection, and tries to find a balance between them. In experiments, we observe that the clipping threshold will become universally smaller when the bit-width is reduced to guarantee sufficient resolution, which further validates the efficaciousness of the gradient in RCF.

## 2.4 WEIGHT NORMALIZATION

In practice, we find that learning $\alpha$ for weights is quite arduous because the distribution of weights is pretty steep and changes frequently during training. As a result, jointly training the clipping threshold and weights parameters is hard to converge. Inspired by the crucial role of batch normalization (BN) (Ioffe & Szegedy, 2015) in activation quantization (Cai et al., 2017), we propose weight normalization (WN) to refine the distribution of weights with zero mean and unit variance,

$$\tilde{\mathcal{W}} = \frac{\mathcal{W} - \mu}{\sigma + \epsilon}, \text{ where } \mu = \frac{1}{I} \sum_{i=1}^{I} \mathcal{W}_i, \sigma = \sqrt{\frac{1}{I} \sum_{i=1}^{I} (\mathcal{W}_i - \mu)^2}, \quad (9)$$

where $\epsilon$ is a small number (typically $10^{-5}$) for numerical stability, and $I$ denotes the number of weights in one layer. Note that quantization of weights is applied right after this normalization.

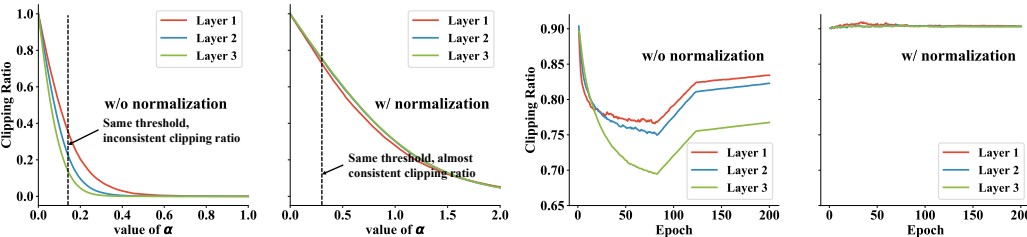

(a) Evolution of clipping ratio with fixed weights  (b) Evolution of clipping ratio with fixed threshold

Figure 4: The evolution of clipping ratio of the first three layers in ResNet-20. (a) demonstrates clipping ratio is too sensitive to threshold to hurt its optimization without weights normalization. (b) shows that weights distribution after normalization is relatively more stable during training.

---

**Algorithm 1** Forward and backward procedure for an APoT quantized convolutional layer

---

**Input:** input activations $\mathcal{X}_{in}$, the full precision weight tensor $\mathcal{W}$, the clipping threshold for weights and activations $\alpha_{\mathcal{W}}, \alpha_{\mathcal{X}}$, the bit-width $b$ of quantized tensor.

**Output:** the output activations $\mathcal{X}_{out}$

1: Normalize weights $\mathcal{W}$ to $\tilde{\mathcal{W}}$
2: Apply RCF and APoT quantization to the normalized weights $\hat{\mathcal{W}} = \alpha_{\mathcal{W}} \Pi_{\mathcal{Q}^a(1,b)} \lfloor \frac{\tilde{\mathcal{W}}}{\alpha_{\mathcal{W}}}, 1 \rceil$
3: Apply RCF and APoT quantization to the activations $\hat{\mathcal{X}_{in}} = \alpha_{\mathcal{X}} \Pi_{\mathcal{Q}^a(1,b)} \lfloor \frac{\mathcal{X}_{in}}{\alpha_{\mathcal{X}}}, 1 \rceil$
4: Compute the output activations $\mathcal{X}_{out} = Conv(\hat{\mathcal{W}}, \hat{\mathcal{X}_{in}})$
5: Compute the loss $\mathcal{L}$ and the gradients $\frac{\partial \mathcal{L}}{\partial \mathcal{X}_{out}}$,
6: Compute the gradients of convolution $\frac{\partial \mathcal{L}}{\partial \hat{\mathcal{X}_{in}}}, \frac{\partial \mathcal{L}}{\partial \hat{\mathcal{W}}}$
7: Compute the gradients for clipping threshold $\frac{\partial \mathcal{L}}{\partial \alpha_{\mathcal{W}}}, \frac{\partial \mathcal{L}}{\partial \alpha_{\mathcal{X}}}$ based on Equation (8)
8: Compute the gradients to the full precision weights $\frac{\partial \mathcal{L}}{\partial \mathcal{W}} = \frac{\partial \mathcal{L}}{\partial \hat{\mathcal{W}}} \frac{\partial \hat{\mathcal{W}}}{\partial \tilde{\mathcal{W}}} \frac{\partial \tilde{\mathcal{W}}}{\partial \mathcal{W}}$
9: Update $\mathcal{W}, \alpha_{\mathcal{W}}, \alpha_{\mathcal{X}}$ with learning rate $\eta_{\mathcal{W}}, \eta_{\alpha_{\mathcal{W}}}, \eta_{\alpha_{\mathcal{X}}}$

---

Normalization is important to provide a relatively consistent and stable input distribution to both clipping and projection functions for smoother optimization of $\alpha$ over different layers and iterations during training. Besides, making the mean of weights to be zero can reap the benefits of the symmetric design of the quantization levels.

Here, we conduct a case study of ResNet-20 on CIFAR10 to illustrate how normalization for weights can help quantization. For a certain layer (at a certain training step) in ResNet-18, We firstly fix the weights, and let $\alpha$ go from 0 to $\max|\mathcal{W}|$ to plot the curve of the clipping ratio (i.e. the proportion of clipped weights). As shown in Figure 4a, the change of clipping ratio is much smoother after quantization. As a result, the optimization of $\alpha$ will be significantly smoother. In addition, normalization also makes the distribution of weights quite more consistent over training iterations. We fix the value of $\alpha$, and visualize clipping ratio over training iterations in Figure 4b. After normalization, the same $\alpha$ will result in almost the same clipping ratio, which improves the consistency of optimization goal for $\alpha$. More experimental analysis demonstrating the effectiveness of the normalization on weights can be found in Appendix B.

## 2.5 TRAINING AND DEPLOYING

We adopt APoT quantization for both weights and activations. Notwithstanding the effect in activations is not conspicuous, we adopt APoT quantization for consistency. During backpropagation, we use STE when computing the gradients of weights, i.e. $\frac{\partial \hat{\mathcal{W}}}{\partial \tilde{\mathcal{W}}} = 1$. The detailed training procedure is shown in Algorithm 1.

To save memory cost during inference, we discard the full precision weights $\mathcal{W}$ and only store the quantized weights $\hat{\mathcal{W}}$. Compared with other uniform quantization methods, APoT quantization is more efficient and effective during inference.

## 3 RELATED WORKS

**Non-Uniform Quantization.** Several methods are proposed for the non-uniform distribution of weights. LQ-Nets(Zhang et al., 2018) learns quantization levels based on the quantization error minimization (QEM) algorithm. Distillation (Polino et al., 2018) optimizes the quantization levels directly to minimize the task loss which reflects the behavior of their teacher network. These methods use finite floating-point numbers to quantize weights (and activations), bringing extra computation overhead compared with linear quantization. Logarithmic quantizers (Zhou et al., 2017; Miyashita et al., 2016) leverage powers-of-2 values to accelerate the computation by *shift* operations; however, they suffer from the *rigid resolution* problem.

**Jointly Training.** Many works have explored to optimize the quantization parameters (e.g., $\alpha$) and the weights parameters simultaneously. Zhu et al. (2016) learns positive and negative scaling coefficients respectively. LQ-Nets jointly train these parameters to minimize the quantization error. QIL (Jung et al., 2019) introduces a learnable transformer to change the quantization intervals and optimize them based on the task loss. PACT (Choi et al., 2018b) parameterizes the clipping threshold in activations and optimize it through gradient descent. However, in PACT, the gradient of $\alpha$ is not accurate, which only includes the contribution from outliers and ignores the contribution from other weights.

**Weight Normalization.** Previous works on weight normalization mainly focus on addressing the limitations of BatchNorm (Ioffe & Szegedy, 2015). Salimans & Kingma (2016); Hoffer et al. (2018) decouple direction from magnitude to accelerate the training procedure. Weight Standardization (Qiao et al., 2019) normalizes weights to zero mean and unit variance during the forward pass. However, there is limited literature that studies the normalization of weights for neural network quantization. (Zhu et al., 2016) uses feature-scaling to normalize weights by dividing the maximum absolute value. Weight Normalization based Quantization (Cai & Li, 2019) also uses this feature scaling and derive the gradient to eliminate the outliers in the weights tensor.

## 4 EXPERIMENT

In this section, we validate our proposed method on ImageNet-ILSVRC2012 (Russakovsky et al., 2015) and CIFAR10 (Krizhevsky et al., 2009). We also conduct ablation study for each component of our algorithm.

### 4.1 EVALUATION ON IMAGENET

We compare our methods with several strong baselines on ResNet architectures (He et al., 2016), including ABC-Net (Lin et al., 2017), DoReFa-Net (Zhou et al., 2016), PACT (Choi et al., 2018b), LQ-Net (Zhang et al., 2018), DSQ (Gong et al., 2019), QIL (Jung et al., 2019). Both weights and activations of the networks are quantized for comparisons. All the state-of-the-art methods use full precision (32 bits) for the first and the last layer, which incur more memory cost. In our implementation, we employ 8-bit quantization for them to balance the accuracy drop and the hardware overhead.

For our proposed APoT quantization algorithm, four configurations of the bit-width, i.e., 2,3,4, and 5 ($k = 2$ and $n = 1$ or $2$ in Equation (5) and (6)) are tested, where one bit is used for the sign of the weights but not for activations. Note that for the 2-bit symmetric weight quantization method, $\mathcal{Q}(\alpha, 2)$ can only be $\{\pm\alpha, 0\}$, therefore only RCF and WN are used in this setting. To obtain a reasonable initialization, we follow Lin et al. (2017); Jung et al. (2019) to initialize our model. Specifically, the 5-bit quantized model is initialized from the pre-trained full precision one[3], while the 4-bit network is initialized from the trained 5-bit model. We compare the accuracy, memory cost, and the fixed point operations under different bit-width. To compare the operations with different bit-width, we use the bit-op computation scheme introduced in Zhou et al. (2016) where the multiplication between a $m$-bit and a $l$-bit uniform quantized number costs $ml$ binary operation. We define one FixOP as one operation between an 8-bit weight and an 8-bit activation which takes 64 binary operations if uniform quantization scheme is applied. In APoT scheme, the multiplication

---

[3]https://pytorch.org/docs/stable/_modules/torchvision/models/resnet.html

Table 1: Comparison of accuracy performance as well as hardware performance of ResNets (He et al., 2016) on ImageNet with existing methods.

| METHOD | PRECISION (W / A) | ACCURACY(%) Top-1 | ACCURACY(%) Top-5 | MODEL SIZE | FixOPS | PRECISION (W / A) | ACCURACY(%) Top-1 | ACCURACY(%) Top-5 | MODEL SIZE | FixOPS |
|---|---|---|---|---|---|---|---|---|---|---|
| FP.(Res18) | 32 / 32 | 70.2 | 89.4 | 46.8 MB | 1.82G | | | | | |
| ABC-Nets | 5 / 5 | 65.0 | 85.9 | 8.72 MB | 781M | | | | | |
| DoReFa-Net | 5 / 5 | 68.4 | 88.3 | 8.72 MB | 781M | 4 / 4 | 68.1 | 88.1 | 7.39 MB | 542M |
| PACT | 5 / 5 | 69.8 | 89.3 | 8.72 MB | 781M | 4 / 4 | 69.2 | 89.0 | 7.39 MB | 542M |
| LQ-Net | | | | | | 4 / 4 | 69.3 | 88.8 | 7.39 MB | 542M |
| DSQ | | | | | | 4 / 4 | 69.6 | - | 7.39 MB | 542M |
| QIL | 5 / 5 | 70.4 | - | 8.72 MB | 781M | 4 / 4 | 70.1 | - | 7.39 MB | 542M |
| **APoT (Ours)** | 5 / 5 | **70.9** | **89.7** | **7.22 MB** | **616M** | 4 / 4 | **70.7** | **89.6** | **5.89 MB** | **437M** |
| ABC-Nets | 3 / 3 | 61.0 | 83.2 | 6.06 MB | 357M | | | | | |
| DoReFa-Net | 3 / 3 | 67.5 | 87.6 | 6.06 MB | 357M | 2 / 2 | 62.6 | 84.6 | 4.73 MB | 225M |
| PACT | 3 / 3 | 68.1 | 88.2 | 6.06 MB | 357M | 2 / 2 | 64.4 | 85.6 | 4.73 MB | 225M |
| LQ-Net | 3 / 3 | 68.2 | 87.9 | 6.06 MB | 357M | 2 / 2 | 64.9 | 85.9 | 4.73 MB | 225M |
| DSQ | 3 / 3 | 68.7 | - | 6.06 MB | 357M | 2 / 2 | 65.2 | - | 4.73 MB | 225M |
| QIL | 3 / 3 | 69.2 | - | 6.06 MB | 357M | 2 / 2 | 65.7 | - | 4.73 MB | 225M |
| PACT+SAWB | | | | | | 2 / 2 | 67.0 | - | 5.36 MB | 243M |
| **APoT (Ours)** | 3 / 3 | **69.9** | **89.2** | **4.56 MB** | **298M** | 2 / 2 | **67.3** | **87.5** | **3.23 MB** | **198M** |
| FP.(Res34) | 32 / 32 | 73.7 | 91.3 | 83.2 MB | 3.68G | | | | | |
| ABC-Nets | 5 / 5 | 68.4 | 88.2 | 14.8 MB | 1.50G | | | | | |
| DSQ | | | | | | 4 / 4 | 72.8 | - | 12.3 MB | 1.00G |
| QIL | 5 / 5 | 73.7 | - | 14.8 MB | 1.50G | 4 / 4 | 73.7 | - | 12.3 MB | 1.00G |
| **APoT (Ours)** | 5 / 5 | **73.9** | **91.6** | **13.3 MB** | **1.15G** | 4 / 4 | **73.8** | **91.6** | **10.8 MB** | **784M** |
| ABC-Nets | 3 / 3 | 66.4 | 87.4 | 9.73 MB | 618M | | | | | |
| LQ-Net | 3 / 3 | 71.9 | 90.2 | 9.73 MB | 618M | 2 / 2 | 69.8 | 89.1 | 7.20 MB | 340M |
| DSQ | 3 / 3 | 72.5 | - | 9.73 MB | 618M | 2 / 2 | 70.0 | - | 7.20 MB | 340M |
| QIL | 3 / 3 | 73.1 | - | 9.73 MB | 618M | 2 / 2 | 70.6 | - | 7.20 MB | 340M |
| **APoT (Ours)** | 3 / 3 | **73.4** | **91.1** | **8.23 MB** | **493M** | 2 / 2 | **70.9** | **89.7** | **5.70 MB** | **285M** |
| FP.(Res50) | 32 / 32 | 76.4 | 93.1 | 97.5 MB | 4.14G | | | | | |
| ABC-Nets | 5 / 5 | 70.1 | 89.7 | 22.2 MB | 1.67G | | | | | |
| DoReFa-Net | 5 / 5 | 71.4 | 93.3 | 22.2 MB | 1.67G | 4 / 4 | 71.4 | 89.8 | 19.4 MB | 1.11G |
| LQ-Net | | | | | | 4 / 4 | 75.1 | 92.4 | 19.4 MB | 1.11G |
| PACT | 5 / 5 | 76.7 | 93.3 | 22.2 MB | 1.67G | 4 / 4 | 76.5 | **93.3** | 19.4 MB | 1.11G |
| **APoT (Ours)** | 5 / 5 | **76.7** | **93.3** | **16.3 MB** | **1.28G** | 4 / 4 | **76.6** | 93.1 | **13.6 MB** | **866M** |
| DoReFa-Net | 3 / 3 | 69.9 | 89.2 | 16.6 MB | 680M | 2 / 2 | 67.1 | 87.3 | 13.8 MB | 370M |
| PACT | 3 / 3 | 75.3 | 92.6 | 16.6 MB | 680M | 2 / 2 | 72.2 | 90.5 | 13.8 MB | 370M |
| LQ-Net | 3 / 3 | 74.2 | 91.6 | 16.6 MB | 680M | 2 / 2 | 71.5 | 90.3 | 13.8 MB | 370M |
| PACT+SAWB | | | | | | 2 / 2 | **74.2** | - | 23.7 MB | 707M |
| **APoT (Ours)** | 3 / 3 | **75.8** | **92.7** | **10.8 MB** | **540M** | 2 / 2 | 73.4 | **91.4** | **7.96 MB** | **308M** |

between a $m$-bit activation and a $l = kn$-bit weight only needs $mn$ shift-adds operations, i.e., $\frac{n \times m}{64}$ FixOPs. More details of the implementation are in the Appendix C.2.

Overall results are shown in Table 1. The results of DoReFa-Net are taken from Choi et al. (2018b), and the other results are quoted from the original papers. It can be observed that our 5-bit quantized network achieves even higher accuracy than the full precision baselines (0.7% Top-1 improvement on ResNet-18 and 0.2% Top-1 improvement on ResNet-34 and ResNet-50), which means quantization may serve the purpose of regularization. Along with the accuracy performance, our APoT quantization can achieve better hardware performance on model size and inference speed. For full precision models, the number in the column of FixOPs indicates FLOPs.

4-bit and 3-bit quantized networks are also preserving (or approaching) the full-precision accuracy except for the 3-bit quantized ResNet-18 and ResNet-34, which only drops 0.5% and 0.3% accuracy respectively. When $b$ is further reduced to 2, our model still outperforms the baselines, which demonstrates the effectiveness of RCF and WN. Note that Choi et al. (2018a) use a full precision shortcut in the model, reaching higher accuracy on ResNet-50 however suffering from the hardware performance. In specific, the different precision between the main path and the residual path may result in greater latency in a pipelined implementation.

## 4.2 EVALUATION ON CIFAR10

We quantize ResNet-20 and ResNet-56 (He et al., 2016) on CIFAR10 for evaluation. We adopt progressive initialization and choose the quantization bit as 2, 3 and 4. More implementations can be found in the Appendix C.2.

Table 2: Accuracy comparison of ResNet architectures on CIFAR10

| MODELS | METHODS | ACCURACY(%) | | |
|---|---|---|---|---|
| | | 2 BITS | 3 BITS | 4 BITS |
| RESNET-20 (FP: 91.6) | DOREFA-NET (ZHOU ET AL., 2016) | 88.2 | 89.9 | 90.5 |
| | PACT (CHOI ET AL., 2018B) | 89.7 | 91.1 | 91.7 |
| | LQ-NET (ZHANG ET AL., 2018) | 90.2 | 91.6 | - |
| | PACT+SAWB+FPSC (CHOI ET AL., 2018A) | 90.5 | - | - |
| | **APOT QUANTIZATION (OURS)** | **91.0** | **92.2** | **92.3** |
| RESNET-56 (FP: 93.2) | PACT+SAWB+FPSC (CHOI ET AL., 2018A) | 92.5 | - | - |
| | **APOT QUANTIZATION (OURS)** | **92.9** | **93.9** | **94.0** |

Table. 2 summarizes the accuracy of our APoT in comparison with baselines. For 3-bit and 4-bit models, APoT quantization has reached comparable results with the full precision baselines. It is worthwhile to note that all state-of-the-arts methods in the table use 4 levels to quantize weights into 2-bit. Our model only employs ternary weights for 2-bit representation and still outstrips existing quantization methods.

## 4.3 ABLATION STUDY

Table 3: Comparison of quantizer, weight normalization and RCF of ResNet-18 on ImageNet.

| METHOD | PRECISION | WN | RCF | ACC.-1 | RCF | ACC.-1 | MODEL SIZE | FIXOPS |
|---|---|---|---|---|---|---|---|---|
| FULL PREC. | 32 / 32 | - | - | 70.2 | - | 70.2 | 46.8 MB | 1.82G |
| **APOT** | 5 / 5 | ✓ | ✓ | 70.9 | ✗ | 70.0 | 7.22 MB | 616M |
| POT | 5 / 5 | ✓ | ✓ | 70.3 | ✗ | 68.9 | 7.22 MB | 582M |
| UNIFORM | 5 / 5 | ✓ | ✓ | 70.7 | ✗ | 69.4 | 7.22 MB | 781M |
| LLOYD | 5 / 5 | ✓ | ✓ | 70.9 | ✗ | 70.2 | 7.22 MB | 1.81G |
| **APOT** | 3 / 3 | ✓ | ✓ | 69.9 | ✗ | 68.5 | 4.56 MB | 298M |
| UNIFORM | 3 / 3 | ✓ | ✓ | 69.4 | ✗ | 67.8 | 4.56 MB | 357M |
| LLOYD | 3 / 3 | ✓ | ✓ | 70.0 | ✗ | 69.0 | 4.56 MB | 1.81G |
| APOT | 3 / 3 | ✗ | ✓ | 2.0 | ✗ | 68.5 | 4.56 MB | 198M |

The proposed algorithm consists of three techniques, APoT quantization levels to fit the bell-shaped distribution, RCF to learn the clipping threshold and WN to avoid the perturbation of the distribution of weights during training. In this section, we conduct an ablation study for these three techniques. We compare the APoT quantizer, the vanilla PoT quantizer, uniform quantizer and a non-uniform quantizer using Lloyd algorithm (Cai et al., 2017) to quantize the weights. And we either apply RCF to learn the optimal clipping range or do not clip any weights (i.e. $\alpha = \max|\mathcal{W}|$). Weight Normalization is also adopted or discarded to justify the effectiveness of these techniques.

Table 3 summarizes the results of ResNet-18 using different techniques. Quantizer using Lloyd achieves the highest accuracy, however, the irregular non-uniform quantized weights cannot utilize the fixed point arithmetic to accelerate the inference time. APoT quantization attends to the distribution of weights, which reaches the same accuracy in 5-bit and only decreases 0.2% accuracy in 3-bit quantization compared with Lloyd, and shares a better tradeoff between task performance and hardware performance. We also observe that the vanilla PoT quantization suffers from the *rigid resolution*, and has the lowest accuracy in the 5-bit model.

Clipping range also matters in quantization, the comparison in Table 3 shows that a proper clipping range can help improve the robustness of the network. Especially when the network is quantized to 3-bit, the accuracy will drop significantly because of the quantization interval increases. Applying RCF to learn the optimal clipping range could improve at most 1.6% accuracy. As we mentioned before, normalization of weight is important to learn the clipping range, and the network diverges if RCF is applied without WN. We refer to the Appendix B for more details of weight normalization during training.

## 5    CONCLUSION

In this paper, we have introduced the additive powers-of-two (APoT) quantization algorithm for quantizing weights and activations in neural networks, which typically exhibit a bell-shaped and long-tailed distribution. Each quantization level of APoT is the sum of a set of powers-of-two terms, bringing roughly 2x speed-up in multiplication compared with uniform quantization. The distribution of the quantization levels matches that of the weights and activations better than existing quantization schemes. In addition, we propose to reparameterize the clipping function and normalize the weights to get a more stable and better-defined gradient for optimizing the clipping threshold. We reach state-of-the-art accuracy on ImageNet and CIFAR10 dataset compared to uniform or PoT quantization.

## ACKNOWLEDGEMENT

This work is supported by National University of Singapore FY2017 SUG Grant, and Singapore Ministry of Education Academic Research Fund Tier 3 under MOEs official grant number MOE2017-T3-1-007.

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

# APPENDICES

## A  GRADIENT DERIVATION

In this section, we derive the gradient estimation of PACT (Choi et al., 2018b) along with our proposed Reparameterized Clipping Function and show the distinction of these two estimation.

### A.1  PACT

Equation (1) shows the forward of PACT. In backpropagation, PACT applies the Straight-Through Estimator for the projection operation. In particular, the STE assumes that

$$\frac{\partial \Pi_{\mathcal{Q}} X}{\partial X} = 1, \tag{10}$$

which means the variable before and after projection are treated the same in backpropagation. Therefore, the gradients of $\alpha$ in PACT is computed by:

$$\frac{\partial \hat{\mathcal{W}}}{\partial \alpha} = \frac{\partial \Pi_{\mathcal{Q}(\alpha,b)} \lfloor \mathcal{W}, \alpha \rceil}{\partial \lfloor \mathcal{W}, \alpha \rceil} \frac{\partial \lfloor \mathcal{W}, \alpha \rceil}{\partial \alpha} = \begin{cases} \operatorname{sign}(\mathcal{W}) & \text{if } |\mathcal{W}| > \alpha \\ 0 & \text{if } |\mathcal{W}| \leq \alpha \end{cases}, \tag{11}$$

where the first term is computed by STE and the second term is because the clip operation $\lfloor \cdot, \alpha \rceil$ returns $\operatorname{sign}(\cdot)\alpha$ when $|\cdot| > \alpha$. In this gradient estimation, the effect of $\alpha$ in the levels set $\mathcal{Q}(\alpha, b)$ is ignored by the STE, leading to an inaccurate approximation.

### A.2  RCF

To avoid the elimination of STE, we reparameterize the clipping function so that the output clipping range before projection is settled and the range is re-scaled after the projection. We can define a general formation of RCF by

$$\hat{\mathcal{W}} = \frac{\alpha}{c} \Pi_{\mathcal{Q}(c,b)} \lfloor \frac{c}{\alpha} \mathcal{W}, c \rceil, \tag{12}$$

where $c > 0$ is a constant. This function clips the weights to $[-c, c]$ before projection and re-scaled to $[-\alpha, \alpha]$ after projection. Thus, the backpropagation is given by:

$$\frac{\partial \hat{\mathcal{W}}}{\partial \alpha} = \frac{\partial \alpha}{\partial \alpha} \times \frac{1}{c} \Pi_{\mathcal{Q}(c,b)} \lfloor \frac{c}{\alpha} \mathcal{W}, c \rceil + \frac{\partial \Pi_{\mathcal{Q}(c,b)} \lfloor \frac{c}{\alpha} \mathcal{W}, c \rceil}{\partial \lfloor \frac{c}{\alpha} \mathcal{W}, c \rceil} \frac{\partial \lfloor \frac{c}{\alpha} \mathcal{W}, c \rceil}{\partial \alpha} \times \frac{\alpha}{c} \tag{13a}$$

$$= \begin{cases} \frac{1}{c} \times \operatorname{sign}(\frac{\alpha}{c} \mathcal{W}) \times c + \frac{\alpha}{c} \times 0 & \text{if } |\mathcal{W}| > \alpha \\ \frac{1}{c} \Pi_{\mathcal{Q}(c,b)} \frac{c}{\alpha} \mathcal{W} + \frac{\alpha}{c} \times (-\frac{c}{\alpha^2}) \mathcal{W} & \text{if } |\mathcal{W}| \leq \alpha \end{cases} \tag{13b}$$

$$= \begin{cases} \operatorname{sign}(\frac{\alpha}{c} \mathcal{W}) & \text{if } |\mathcal{W}| > \alpha \\ \frac{1}{c} \Pi_{\mathcal{Q}(c,b)} \frac{c}{\alpha} \mathcal{W} - \frac{1}{\alpha} \mathcal{W} & \text{if } |\mathcal{W}| \leq \alpha \end{cases}. \tag{13c}$$

Since the levels set $\mathcal{Q}$ is not parameterized by $\alpha$, the gradients will flow to two parts in RCF: the re-scale coefficient and the scale in clipping function. The constant $c$ here do not impact the gradient estimation, therefore we choose 1 for simplicity in the implementation. Note that in uniform quantization scheme, this function is equivalent to the Learned Step Size Quantization (Esser et al., 2020), where the setp size is the same for all levels while RCF provides a more general formation for any levels set $\mathcal{Q}$.

## B  HOW DOES NORMALIZATION HELP QUANTIZATION

In this section, we show some experimental results to illustrate the effect of our weights normalization in quantization neural networks.

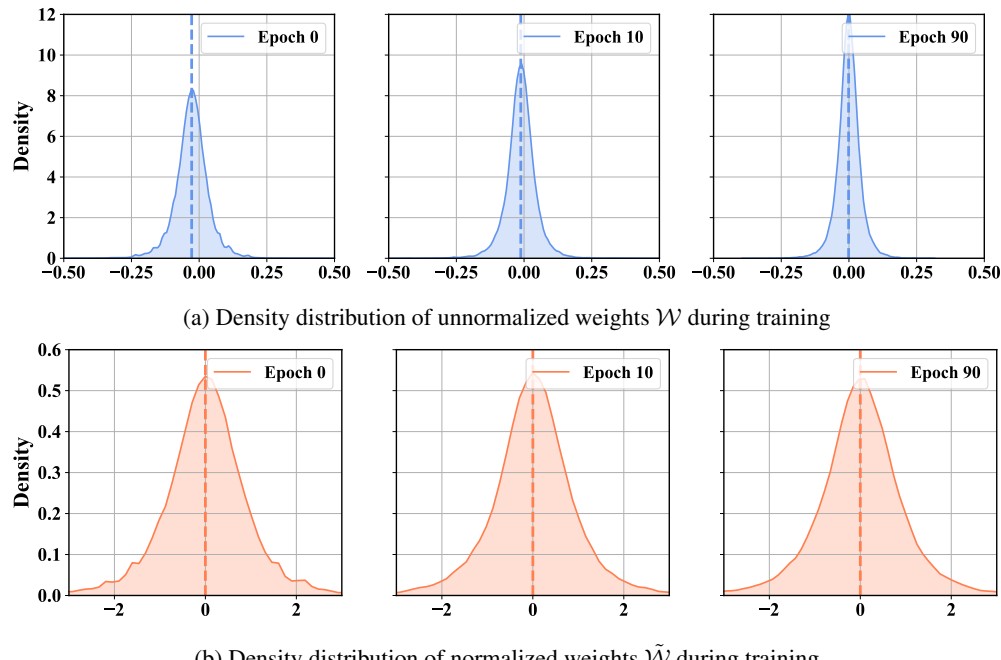

(a) Density distribution of unnormalized weights $\mathcal{W}$ during training

(b) Density distribution of normalized weights $\tilde{\mathcal{W}}$ during training

Figure 5: When weights are normalized the distribution of weights are more stable. The dashed line shows the mean value of weights.

## B.1 WEIGHTS DISTRIBUTION

We visualize the density distribution of weights before normalization $\mathcal{W}$ and after normalization $\tilde{\mathcal{W}}$ during training to demonstrate its effectiveness.

Figure 5a demonstrates the density distribution of the fifth layer of the 5-bit quantized ResNet-18, from which we can see that the density of the unnormalized weights could be extensive high ($> 8$) in the centered area. Such distribution indicates that even a tiny change of clipping threshold would bring a significant effect on clipping when $\alpha$ is small, as shown in Figure 4a. , which means a small learning rate for $\alpha$ is needed. However, if the learning rate is too small, the change of $\alpha$ cannot follow the change of weights distribution because weights are also updated according to Figure 5a. Thus it is unfavorable to train the clipping threshold for unnormalized weights, while Figure 5b shows that the normalized weights can have a stable distribution. Furthermore, the dashed line in the figure indicates $\mathcal{W}$ usually do not have zero mean, which may not utilize the symmetric design of quantization levels.

## B.2 TRAINING BEHAVIOR

The above experiments use normalization during training to compare the distribution of weights. In this section, we compare the training of quantization neural networks with and without normalization to investigate the real effect of WN. Here, we train a 3-bit quantized (full precision for activations) ResNet-20 from scratch, and compare the results under different learning rate for $\alpha$. The results are shown in Table 4, from which we can find that if weights are normalized during training, the network can converge to descent performances and is robust to the learning rate of clipping threshold. However, if the weights are not normalized, the network would diverge if the learning rate for $\alpha$ is too high. Even if the learning rate is set to a lower value, the network does not outperform the normalized one. Based on the training behaviors, the learning rate for clipping threshold without WN in QNNs need a careful choice.

Table 4: Accuracy comparison of 3-bit quantized ResNet-20 on CIFAR10.

| LEARNING RATE | 0.1 | 0.01 | 0.001 | 0.0001 |
|---|---|---|---|---|
| W/ NORMALIZATION | 91.6 | 91.7 | 91.6 | 91.8 |
| W/O NORMALIZATION | 0.2 | 0.2 | 62.8 | 84.7 |

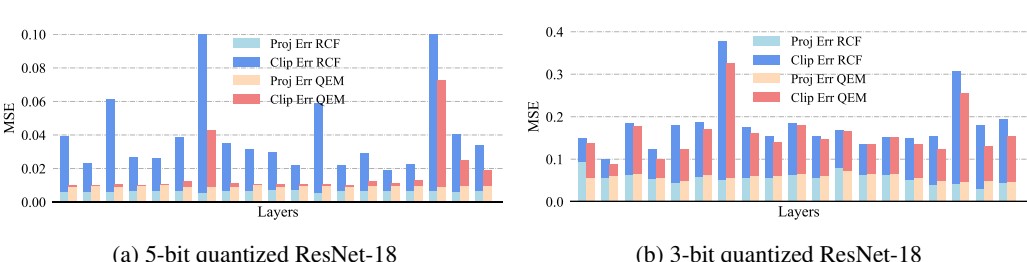

(a) 5-bit quantized ResNet-18        (b) 3-bit quantized ResNet-18

Figure 6: A summary of projection error and clipping error in different layers.

## C   EXPERIMENTAL DETAILS

### C.1   REVISITING QUANTIZATION ERROR

Typically, quantization error ($\Delta$) is defined as the mean squared error between weights $\tilde{\mathcal{W}}$ and $\hat{\mathcal{W}}$ before and after quantization respectively, defined as $\Delta = \mathbb{E}[\tilde{\mathcal{W}} - \hat{\mathcal{W}}]^2$. This quantization error is composed of two errors, the clipping error $\Delta_{clip}$ produced by $\lfloor \cdot, \alpha \rceil$ and the projection error $\Delta_{proj}$ produced by $\Pi_{\mathcal{Q}}$. I.e.

$$\Delta = \Delta_{clip} + \Delta_{proj} = \frac{1}{I} \sum_{|\tilde{\mathcal{W}}_i| > \alpha} \left( |\tilde{\mathcal{W}}_i| - \alpha \right)^2 + \frac{1}{I} \sum_{|\tilde{\mathcal{W}}_i| \leq \alpha} (\tilde{\mathcal{W}}_i - \hat{\mathcal{W}}_i)^2. \tag{14}$$

Previous methods (Zhang et al., 2018; Cai et al., 2017) seek to minimize the quantization error to obtain the optimal clipping threshold (i.e. $\alpha = \arg\min_\alpha (\Delta_{clip} + \Delta_{proj})$), while RCF is directly optimized by the final training loss to balance projection error and clipping error. We compare the Quantization Error Minimization (QEM) method with our RCF on the quantized ResNet-18 model. Figure 6 gives an overview of the clipping error and projection error using RCF or QEM.

For the 5-bit quantized model, RCF has a much higher quantization error. The projection error obtained by RCF is lower than QEM and QEM significantly reduces the clipping error. Therefore, we can infer that projection error has a higher priority in RCF. When quantizing to 3-bit, the clipping error in RCF still exceeds QEM except for the first quantized layer. This means RCF can identify whether the projection is more important than the clipping over different layers and bit-width. Generally, the insight behind is that simply minimizing the quantization error may not be the best choice and it is more direct to optimize threshold with respect to training loss.

### C.2   IMPLEMENTATIONS DETAILS

The ImageNet dataset consists of 1.2M training and 50K validation images. We use a standard data preprocess in the original paper (He et al., 2016). For training images, they are randomly cropped and resized to $224 \times 224$. Validation images are center-cropped to the same size. We use the Pytorch official code [4] to construct ResNets, and they are initialized from the released pre-trained model. We use stochastic gradient descent (SGD) with the momentum of 0.9 to optimize both weight parameters and the clipping threshold simultaneously. Batch size is set to 1024 and the learning rate starts from 0.1 with a decay factor of 0.1 at epoch 30,60,80,100. The network is trained up to 120 epochs and weight decay is set to $10^{-4}$ for 3-bit quantized models or higher and $2 \times 10^{-5}$ for 2-bit model.

The CIFAR10 dataset contains 50K training and 10K test images with $32 \times 32$ pixels. The ResNet architectures for CIFAR10 (He et al., 2016) contains a convolutional layer followed by 3 residual

---

[4] https://github.com/pytorch/vision/blob/master/torchvision/models/resnet.py

blocks and a final FC layer. We train full precision ResNet-20 and ResNet-56 firstly and use them as initialization for quantized models. All networks were trained for 200 epochs with a mini-batch size of 128. SGD with momentum of 0.9 was adopted to optimize the parameters. Learning rate started at 0.04 and was scaled by 0.1 at epoch 80,120. Weight decay was set to $10^{-4}$.

For clipping threshold $\alpha$, we set 8.0 for activations and 3.0 for weights as initial value when training a 5-bit quantized model. The learning rate of $\alpha$ is set to 0.01 and 0.03 for weights and activations, respectively. During practice, we found that the learning rate of $\alpha$ merely does not influence network performance. Different from PACT (Choi et al., 2018b), the update of $\alpha$ in our works already consider the projection error, so we do not require a relatively large L2-regularization. In practice, the network works fine when the weight decay for $\alpha$ is set to $10^{-5}$ and may increase to $10^{-4}$ when bit-width is reduced.

