# OpenReview forum: "Additive Powers-of-Two Quantization: An Efficient Non-uniform Discretization for Neural Networks"
_ICLR.cc/2020/Conference — Accept (Poster)_

### Official Review · AnonReviewer3 · 2019-10-18
**Official Blind Review #3**

**Rating:** 6

**Review:**

Summary:
The authors propose to compress Neural Networks (NNs) by quantizing their weights to sums of powers-of-two. This allows both to take the non-uniform distribution of the weights into account and to perform fast inference on dedicated hardware.

Strengths of the paper:
- The problem is clearly stated (in particular the two questions about the clipping operation and the quantization levels are clearly presented in the Introduction). Similarly, the authors alleviate what they call the "rigid resolution" problem by using sums of powers-of-two and clearly explain their choice.
- The paper addresses the inference time, which is an important metric. Indeed, researchers tend to focus on the size of the compressed weights as this is easily measurable and not questionable. Inference time however depends on the hardware but is crucial as compressed models often that run on embedded devices are often required to run in real-time (say image detection models). The paragraph "Computation" in Section 2.1 is therefore useful for the reader.

Weaknesses of the paper:
- Both the reparameterization of the clipping function and the weight normalization before quantization approaches seem not novel to me, see for instance: "Weight Normalization based Quantization for Deep Neural Network Compression", Cai et al.
- The experiments are lacking the widely used ResNet-50 baseline (and Table 4 should be in the main paper instead of the appendix). Other tasks such as Image Detection (Mask R-CNN) could also strengthen the impact of the paper.
- Moreover, unless I am mistaken, the authors "do not quantize the first and last layers". While not quantizing the first layer has no impact on the compressed size (weights of size 7x7x3x64 = 36 KB for a ResNet-18), not quantizing the last layer (ie the classifier, weights of size 512x1000 = 1.95 MB) seems problematic. It is indeed challenging to quantize the classifier, however this adds an overhead of 1.95 MB. Assuming the ResNet18 is compressed with 2 bits/weight, the compressed part has a size of  44.6 MB/16 = 2.8 MB. Thus, with the classifier, its size is 2.8+1.95 = 4.75 MB, which is roughly a x10 compression factor.

Justification of rating:
The paper presents an interesting idea that accounts both for the inference time and the bell-shaped distribution of weights in NNs. However, the results must be pushed further (see "weaknesses").


=== After the rebuttal ===

I thank the authors for their rebuttal. They answer to all the raised concerns with clarity and concision and performed additional experiments.

In particular, the authors:
- perform additional ResNet50 experiments
- quantize the input layer and the classifier to 8 bits
- clarifying the positioning and in particular with respect to weight normalization

The paper is substantially better in its updated version. Therefore, I am happy to update my rating to "weak accept".


**Experience Assessment:**

I have published one or two papers in this area.

**Review Assessment: Checking Correctness Of Derivations And Theory:**

N/A

**Review Assessment: Checking Correctness Of Experiments:**

I carefully checked the experiments.

**Review Assessment: Thoroughness In Paper Reading:**

I read the paper thoroughly.

---

> ### Author Response · Authors · 2019-11-08
> **Response to Review #3**
>
> We thank you for your time and thoughtful review. We would like to respond to the weakness and hope we addressed all the concerns. We've also made revisions to the manuscript. We look forward to your feedback.
>
> " Both the reparameterization of the clipping function and the weight normalization before quantization approaches seem not novel to me, see for instance: Weight Normalization based Quantization for Deep Neural Network Compression "
>
> R: In fact, we think Weight Normalization based Quantization (WNQ) and our proposed method are based on different ideas. The weight normalization in WNQ aims to eliminate the outliers in weight tensor, and the range of quantized weights ($\alpha$) is obtained by Quantization Error Minimization (QEM).
> Instead, we learn $\alpha$ by gradient descent w.r.t. the task loss thus clips the outliers. In Section 4.4, we compare this learning-based RCF and rule-based QEM algorithm. Weight normalization here aims to stabilize the distribution of weights during training, not eliminate the outliers. We show that RCF without WN is hard to converge in Section 4.3 and Appendix A. The following table summarizes the difference. We will cite WNQ in our related work, however, our work by no means uses a similar method as WNQ.
>   —————————————————————————————————-
> | Methods |  determining $\alpha$	                          | weights normalization
> | WNQ       | minimize the quantization error | avoid (eliminate) outliers
> | Ours	    | minimize the training objective  | stabilize the weights for RCF
> —————————————————————————————————-
> In the new version, we conduct an ablation study on RCF and WN and different quantizers in Section 4.3, showing that the learning-based RCF can improve the quantization robustness and WN help the convergence of the model.
>
>
> "The experiments are lacking the widely used ResNet-50 baseline (and Table 4 should be in the main paper instead of the appendix). Other tasks such as Image Detection (Mask R-CNN) could also strengthen the impact of the paper. "
>
> R: We have updated the results of the ResNet-50 in Table 3. There are many tasks to evaluate, e.g., detection tasks, image segmentation tasks, etc. However, image classification is the most widely accepted benchmark and all our baselines (of the existing state-of-the-art quantization methods) are using image classification. Consequently, we just follow the existing papers to evaluate the performance of image classification. CNN models with better performance for image classification usually perform better for other tasks, such as [1,2]
>
> "Moreover, unless I am mistaken, the authors "do not quantize the first and last layers". While not quantizing the first layer has no impact on the compressed size (weights of size 7x7x3x64 = 36 KB for a ResNet-18), not quantizing the last layer (ie the classifier, weights of size 512x1000 = 1.95 MB) seems problematic. It is indeed challenging to quantize the classifier, however, this adds an overhead of 1.95 MB. Assuming the ResNet18 is compressed with 2 bits/weight, the compressed part has a size of  44.6 MB/16 = 2.8 MB. Thus, with the classifier, its size is 2.8+1.95 = 4.75 MB, which is roughly an x10 compression factor. "
>
> R: The reviewer noted correctly that the size of the classifier has a huge impact. We would like to clarify the reason we use full precision for the first and last layers in our original paper.
> 1. ALL our baselines have claimed to use the full precision first and last layers, and we are following their configurations to fairly evaluate our method. In fact, if we all (our method and baselines) compress both the first and last layers, our method would still have a smaller size.
> 2. Be that as it may, we agree with the reviewer that using full precision for the first and the last layer is a little bit inefficient. Therefore, we apply 8-bit quantization to them and finetune our trained model with full precision first and last layer for 1 epoch and find no accuracy degradation. When quantizing the last layer to 8-bit, our 2-bit ResNet-18 only has 3.23MB, which means the compression ratio is 14.8 times. However, if we go further with fewer than 8bits, the accuracy would degrade a little. In Table 1-3, we add the model size as well as computation comparison with the existing methods. We highlight that our method has the lowest model size and fixed point operations, which justify the effectiveness and efficiency of APoT quantization.
> 3. Another metric in hardware performance is computation. Though the classifier has a huge model size (1.95 MB in ResNet-18, approximately 4.16% of all model size), the operation (multiplication and accumulation) introduced by the classifier is (512+511+1)x1000=1.024M, which only occupies 1.024M $\div$ 1.81G = 0.057% operations in ResNet-18. Therefore, the latency in the last layer is significantly less than the other layers.
>
> [1] Deep Residual Learning for Image Recognition
> [2] MobileNetV2: Inverted Residuals and Linear Bottlenecks

---

### Official Review · AnonReviewer1 · 2019-10-29
**Official Blind Review #1**

**Rating:** 3

**Review:**


The paper presents an approach based on power-two quantization to compress the weights of neural networks. The authors elaborate on a coding scheme that adds several quantized values with different log scales (typically 2^(-2k) and 2^(-2k+1) for varying k, for the sum of two log scales). More importantly, they emphasize on the importance of weight scaling and learning the clipping threshold. experiments are carried out with several versions of ResNet on CIFAR-10 and ImageNet, and the authors show better performances than competing baseline for a fixed quantization budget per float.

The results are better than the non-compressed baseline (CIFAR with 3 and 5 bit/flotat, ImageNet with 5bit/float.) This seems surprising. Is there any clear reason why? Shouldn't the performance decrease as a result of the compression?

While the results are good, a large part of the paper is dedicated to the idea of summing powers-of-two quantizers. The motivation comes from the sub optimality of uniform quantizers. Some ablation studies demonstrate that the scheme power-of-two is better than uniform. Nonetheless, the results in Table 2 indicate that uniform already achieves very good performances (already better than the non-compressed network for top-1 accuracy), so that most of the gain seems to come from weight normalization and learning the clipping threshold.

Thus, the advantage of the exact scheme that is proposed compared to other methods is unclear to me. There already are well-known algorithms for non-uniform quantization (e.g., Lloyd). Lin et al. ("Towards Accurate Binary Convolutional Neural Network", cited in the paper) already addressed the non-uniform quantization by learning different binary bases. While the results of the paper are better than a number of previous results, it seems to me that most of the gain does not come from the quantizer but rather from the other elements of the method.

The contribution then seems a bit incremental, as it is mostly weight normalization +  straight through estimator to learn the clipping constant by minimizing directly the loss function. One way to make the contribution stronger is to isolate this part to see its effect more generally on several non-uniform quantization schemes.

other comments:
- is there any (possibly intuitive) justification for the fact that the results of the compressed networks are better than the non-compressed ones?


**Experience Assessment:**

I do not know much about this area.

**Review Assessment: Checking Correctness Of Derivations And Theory:**

I assessed the sensibility of the derivations and theory.

**Review Assessment: Checking Correctness Of Experiments:**

I assessed the sensibility of the experiments.

**Review Assessment: Thoroughness In Paper Reading:**

I read the paper at least twice and used my best judgement in assessing the paper.

---

> ### Author Response · Authors · 2019-11-08
> **Response to Review #1**
>
> Thank you for your constructive feedback. We have revised the manuscript to improve the presentation and hope we address your concerns.
>
> "The results are better than the non-compressed baseline (CIFAR with 3 and 5 bit/float, ImageNet with 5bit/float.) This seems surprising. Is there any clear reason why? Shouldn't the performance decrease as a result of the compression? is there any (possibly intuitive) justification for the fact that the results of the compressed networks are better than the non-compressed ones?"
>
> R: In our view, the improvement comes from two aspects. 1) [1] observes that initializing from a pretrained model can help the quantized model converge to a better minimum. We think initializing from a pre-trained model serves the purpose of warm restarts, though the parameters are quantized, can still be helpful. And SGD with warm restarts usually helps to find a better optimum. 2) [2] reported higher accuracy on CIFAR10 and the explanation is that neural networks are over-parameterized and compression could prevent the network from overfitting, thus acts like regularization to improve the neural network performance.
>
>
> "Thus, the advantage of the exact scheme that is proposed compared to other methods is unclear to me. There already are well-known algorithms for non-uniform quantization (e.g., Lloyd). Lin et al. ("Towards Accurate Binary Convolutional Neural Network", cited in the paper) already addressed the non-uniform quantization by learning different binary bases. While the results of the paper are better than a number of previous results, it seems to me that most of the gain does not come from the quantizer but rather from the other elements of the method. "
>
> R: APoT quantization is effective and efficient as well (Faster than ABC-Net, because we have less binary bases in 5-bit). We highlight our HARDWARE performance compared with existing methods. APoT quantizer is not only suitable for the distribution but also reduces the hardware overhead in uniform quantization. There indeed has been some confusion in our previous presentation to show the effectiveness of our APoT quantization and other techniques (WN, RCF). We have submitted a new version of the manuscript to address this confusion. Firstly, we add a figure in Page 4 to illustrates the computation pattern under different k. In experiments, we compare accuracy, model size and computation cost in Table 1-3 to demonstrate the effectiveness and the efficiency of our proposed APoT.
>
>
> “The contribution then seems a bit incremental, as it is mostly weight normalization +  straight-through estimator to learn the clipping constant by minimizing directly the loss function. One way to make the contribution stronger is to isolate this part to see its effect more generally on several non-uniform quantization schemes.”
>
> R: We conduct a thorough ablation study in Section 4.3, in this section, we compare the different quantizers (PoT, uniform, the non-uniform Lloyd), the usage of WN and RCF and report the accuracy as well as the hardware performance. Shown in Table 5, APoT+WN+RCF is able to find a better balance between task performance and hardware cost, all of which are important to obtain higher accuracy and achieve low latency and memory costs. We also show that WN+RCF can improve the performances of other quantizers. (e.g. 1.6% improvement for 3-bit uniform quantization). Please refer to our newly submitted version for more details and share us with your constructive feedback.
>
> [1] Learning to Quantize Deep Networks by Optimizing Quantization Intervals with Task Loss
> [2] Trained Ternary Quantization

---

### Official Review · AnonReviewer2 · 2019-10-30
**Official Blind Review #2**

**Rating:** 6

**Review:**

In this paper, the authors proposed a novel quantization method, additive powers-of-two quantization and show both the uniform quantization and powers-of-two quantization are the special case of this method. To train such a quantized network, new clipping function, and weight normalization are proposed. The numerical results show that the proposed method only introduces a small accuracy loss and sometimes even improve accuracy. Compared to the other methods, it also shows better performance. Overall, I think this work is valuable and may be considered for publication. But the reviewer is completely out of this neural network quantization area, and thus not very familiar with the related works.

The following are some more detailed comments:

1. On page 3, "when we increase the bit-width from b to b+1 ... be split into 2^b subintervals." Based on Equation 3, I think it might be 2^(b-1) +1 subintervals.

2. From Figure 1 (c), the introduced quantization method introduces non-monotonic interval steps, which is a little unintuitive. Can the authors explain if this can be further improved?

3. A very important theme of this paper is on hardware, however, I feel the paper doesn't have a satisfying discussion on the hardware implementation. In order to make the argument more solid, the authors may want to provide more discussion on the hardware implementation trade-offs. Also, a detailed comparison of the quantization size comparison should also be provided.

4. The authors claim the quantized network can sometimes achieve better performance. This statement needs to be further checked. Since the performance provided in the baseline Resnet models can be slightly improved with further training and training schedule tuning. Without a thorough optimization, such a claim might be misleading.

**Experience Assessment:**

I do not know much about this area.

**Review Assessment: Checking Correctness Of Derivations And Theory:**

I did not assess the derivations or theory.

**Review Assessment: Checking Correctness Of Experiments:**

I assessed the sensibility of the experiments.

**Review Assessment: Thoroughness In Paper Reading:**

I read the paper at least twice and used my best judgement in assessing the paper.

---

> ### Author Response · Authors · 2019-11-08
> **Response to Review #2**
>
> Thank you for the positive review. We have made changes and submitted a new version to address the typos or some minor technical errors, please check it.
>
> “1. On page 3, "when we increase the bit-width from b to b+1 ... be split into 2^b subintervals." Based on Equation 3, I think it might be 2^(b-1) +1 subintervals.”
> R:  We have fixed it in the updated version. Thank you for pointing it out!
>
>
> "From Figure 1 (c), the introduced quantization method introduces non-monotonic interval steps, which is a little unintuitive. Can the authors explain if this can be further improved?"
> R: APoT constrains the quantization levels as a sum of PoT terms, therefore inevitably introduces non-monotonic steps when a small PoT term adds to a large PoT term. While PoT quantization uses monotonic steps (multiplied by 2), we have shown its rigid resolution problem. As for some other non-uniform quantization (e.g. Lloyd) that might have monotonic steps, they could not utilize the hardware accelerator. APoT leverages the computation and accuracy performances. Please see Section 4.3 Ablation Study for a comparison of these different quantizers.
> There is a possible way to avoid this non-monotonic steps by only adding two large terms. However, the definition in Equation 5 must change, and $p_i$ will have different possible values when it is added to different PoT terms. As a result, this definition cannot generalize the uniform case and PoT case and make decoding of the bit representations complexed. We will leave this as future work to find more potential combinations between PoT terms.
>
>
> "A very important theme of this paper is on hardware, however, I feel the paper doesn't have a satisfying discussion on the hardware implementation. To make the argument more solid, the authors may want to provide more discussion on the hardware implementation trade-offs. Also, a detailed comparison of the quantization size comparison should also be provided."
> R: Thanks for the advice about this work. We provided more discussion in the newly submitted version. We added a figure and more experiment details to show the hardware performances. The results show that our APoT quantization can significantly reduce the fixed point operations and improve the accuracy of uniform along with the PoT quantization.
>
>
> "The authors claim the quantized network can sometimes achieve better performance. This statement needs to be further checked. Since the performance provided in the baseline Resnet models can be slightly improved with further training and training schedule tuning. Without a thorough optimization, such a claim might be misleading."
> R: We agree with the reviewer that full precision models can be improved further. We will revise this statement.

---

### Author Response · Authors · 2019-11-08
**Submission Updated**

We thank all reviewers for their time and valuable comments. We have revised the paper following the reviewers’ suggestions to further improve the presentation and updated the submission. The revisions we made include the following:

1) We added a figure in Page 4 to illustrate the computation pattern with different $k$, which shows that APoT is more efficient than the uniform quantization to perform convolution.
2) In Section 4.1, we replaced the accuracy bar graph with 3 Tables to show the accuracy and hardware comparison (model size and fixed point ops) to justify the effectiveness and the efficiency of ours and existing methods. We added the experiments of ResNet-50 in Table 3 as suggested by reviewer #3.
3) In Section 4.3, we conducted an ablation study to evaluate the three techniques (APoT, RCF, WN) introduced in this work.
4) Some notations are changed to improve the presentations and make statements clearer.

---

> ### Author Response · Authors · 2020-01-01
> **Update for the Final Version**
>
> We thank all reviewers and area chairs for their kind reviews and support of the acceptance for this paper. The final version of the manuscript is uploaded, the change we made includes:
>
> 1. We change some figures to improve clarity and add more information. A new figure illustrating the distribution of weights is added in the introduction.
> 2. APoT quantization is now applied to activations and also capable to generalize 2n+1 bitwidths.
> 3. The link for training code is provided and the code will be available soon.

---

### Public Comment · ~Thomas_Pfeil1 · 2020-04-17
**Reference using similar methods**

Please see the following publication for similar methods. However, here, additive powers-of-two quantization is not applied during, but after training.

S. Vogel, J. Springer, A. Guntoro and G. Ascheid, "Self-Supervised Quantization of Pre-Trained Neural Networks for Multiplierless Acceleration," 2019 Design, Automation & Test in Europe Conference & Exhibition (DATE), Florence, Italy, 2019, pp. 1094-1099.

---

### Decision · Program_Chairs · 2019-12-19

**Decision:**

Accept (Poster)

**Comment:**

This paper presents a quantization scheme with the advantage of high computational efﬁciency. The experimental results show that the proposed scheme outperforms SOTA methods and is competitive with the full-precision models. The reviewers initially raised some concerns including baseline ResNet performance,  detailed comparison of the quantization size, and comparison with ResNet50. Authors addressed these concerns in the rebuttal and revised the draft to accommodate the requested items. The reviewers appreciated the revision and find it highly improved. Their overall recommendation is toward accept, which I also support.